# The Characterization of a Novel Virus Discovered in the Yeast *Pichia membranifaciens*

**DOI:** 10.3390/v14030594

**Published:** 2022-03-13

**Authors:** Mark D. Lee, Jack W. Creagh, Lance R. Fredericks, Angela M. Crabtree, Jagdish Suresh Patel, Paul A. Rowley

**Affiliations:** 1Department of Biological Sciences, University of Idaho, Moscow, ID 83844, USA; leemark518@gmail.com (M.D.L.); crea4321@vandals.uidaho.edu (J.W.C.); fred6557@vandals.uidaho.edu (L.R.F.); angela.crabtree.88@gmail.com (A.M.C.); jpatel@uidaho.edu (J.S.P.); 2Center for Modeling Complex Interactions, University of Idaho, Moscow, ID 83844, USA

**Keywords:** Pichia, totivirus, yeast, molecular modeling, double-stranded RNA

## Abstract

Mycoviruses are widely distributed across fungi, including the yeasts of the Saccharomycotina subphylum. This manuscript reports the first double-stranded RNA (dsRNA) virus isolated from *Pichia membranifaciens.* This novel virus has been named Pichia membranifaciens virus L-A (PmV-L-A) and is a member of the *Totiviridae*. PmV-L-A is 4579 bp in length, with RNA secondary structures similar to the packaging, replication, and frameshift signals of totiviruses that infect Saccharomycotina yeasts. PmV-L-A was found to be part of a monophyletic group within the I-A totiviruses, implying a shared ancestry between mycoviruses isolated from the *Pichiaceae* and *Saccharomycetaceae* yeasts. Energy-minimized AlphaFold2 molecular models of the PmV-L-A Gag protein revealed structural conservation with the Gag protein of Saccharomyces cerevisiae virus L-A (ScV-L-A). The predicted tertiary structure of the PmV-L-A Pol and other homologs provided a possible mechanism for totivirus RNA replication due to structural similarities with the RNA-dependent RNA polymerases of mammalian dsRNA viruses. Insights into the structure, function, and evolution of totiviruses gained from yeasts are essential because of their emerging role in animal disease and their parallels with mammalian viruses.

## 1. Introduction

*Pichia membranifaciens* is a pseudofilamentous fungus and a common spoilage organism that contaminates meats, soft cheeses, vegetables, and beverages [1]. Although *P. membranifaciens* is frequently associated with spoilage, it can act as a biocontrol agent against plant pathogens and aggressive post-harvest spoilage organisms [2,3,4,5,6,7,8]. Biological control by *Pichia* species relies on volatile organic compounds, killer toxins, and glucanases [9,10,11,12]. Killer toxin production by yeast is often associated with double-stranded RNA (dsRNA) viruses and dsRNA satellites. Previous attempts to discover novel dsRNA-associated killer toxins in *P. membranifaciens* found that the strain NCYC333 plays host to a putative dsRNA mycovirus, most likely a member of the *Totiviridae* family [9,13].

The number of known mycoviruses has expanded over recent years, with the International Committee for the Taxonomy of Viruses (ICTV) recognizing mycoviruses with RNA and DNA genomes [14]. Mycoviruses can be found across all major taxonomic groups of fungi and are not thought to spread via extracellular transmission but instead transmit vertically by mitosis and meiosis, and horizontally due to anastomosis or yeast mating. Generally, mycovirus infections are considered benign, with little obvious fitness cost to the host. However, there are instances where mycovirus infection results in noticeable phenotypes, such as the alteration of fungal virulence, morphology, growth rate, and pigmentation, so mycoviruses have been used to control fungal diseases [15,16]. There are descriptions of viruses from the *Saccharomycetaceae* yeasts but no published descriptions of totiviruses in the *Pichiaceae*. Given that mycovirus infection can alter host physiology, and *P. membranifaciens* holds future promise as a biological control agent, it is crucial to characterize the viruses associated with the species. 

Mycoviruses in the family *Totiviridae* are dsRNA viruses that are distributed across diverse fungal species (*Totivirus* and *Victorivirus* genera) but are also found in numerous species of human and animal pathogenic protozoa (*Giardiavirus*, *Trichomonasvirus*, and *Leishmaniavirus* genera) [17,18,19,20,21]. In addition, there are reports of totivirus-like viruses that infect members of the phylum Arthropoda, specifically shrimps, and mosquitos [22,23]. A well-studied member of the *Totiviridae* family is the L-A virus of the brewer’s/baker’s yeast *Saccharomyces cerevisiae* (Saccharomyces cerevisiae virus L-A; ScV-L-A) [20,24]. The discovery of ScV-L-A led to the description of many other dsRNA viruses within the yeasts of the *Saccharomycetaceae*, including *Saccharomyces paradoxus*, *Saccharomyces uvarum*, *Saccharomyces kudriavzevii*, and *Torulaspora delbrueckii* [25,26,27]. These viruses typically replicate within the cytosol and have a dsRNA genome of 4580 bp that encodes two proteins, Gag and Gag-pol. The Gag protein encapsidates the viral genome and binds host mRNA 5′ caps that are then ligated to viral transcripts [28]. Translation of Gag-pol requires a stem-loop and slippery site to induce ribosome stalling and −1 frameshifting [29]. The Gag-pol fusion enables the tethering of the Pol domain to the inner surface of the viral capsid (1–2 per capsid). The Pol domain of totiviruses is an RNA-dependent RNA polymerase (RdRP) of unknown structure responsible for the synthesis of viral RNAs. Pol is essential for (+) sense RNA packaging and the subsequent synthesis of complementary (−) sense RNAs to form the dsRNA genome [30,31]. The genomic dsRNA is used as a template for conservative RNA replication to produce (+) sense viral transcripts that are extruded to the cytosol for capping, translation, and packaging into new viral particles. 

This manuscript describes the discovery and characterization of a novel member of the *Totiviridae* isolated from *P. membranifaciens.* As the first example of a dsRNA virus in the species, we have named it Pichia membranifaciens virus L-A (PmV-L-A), consistent with the naming of the first virus discovered in *S. cerevisiae*. The organization of the PmV-L-A genome is the same as other related yeast totiviruses. It includes conserved RNA structures that are likely essential for RNA frameshifting, replication, and packaging. The nucleotide and protein divergence classifies this virus as a new species and is an outgroup to other viruses of the *Saccharomycetaceae*. Structural models of the PmV-L-A proteins are consistent with known structures of totivirus capsid proteins and RdRP proteins of dsRNA viruses that infect mammals. 

## 2. Methods

### 2.1. Purification and Digestion of dsRNAs with RNase III

DsRNAs were purified from the *P. membranifaciens* strain NCYC333 (obtained from the National Collection of Yeast Cultures, Norwich, UK) using the method described previously [13]. Five µg of dsRNAs were incubated with ten units of ShortCut^®^ RNase III (New England Biolabs, Ipswich, MA, USA; catalog #M0245) in the buffer conditions recommended by the manufacturer for 20 min at 37 °C. The reaction was stopped by adding EDTA to a final concentration of 45 mM. DNase I (New England Biolabs, Ipswich, MA, USA; catalog # M0303S) digestion was performed as directed by the manufacturer in 1 × reaction buffer for 10 min at 37 °C. The reaction was stopped by incubating at 75 °C for 10 min. All digested products were analyzed by agarose gel electrophoresis with ethidium bromide staining.

### 2.2. Determining the Genetic Sequence of dsRNAs

All methods for the sequencing of dsRNAs were as previously described [13]. Briefly, 5′ rapid amplification of cDNA ends (RACE) was carried out as defined in the manufacturer’s instructions, using purified dsRNAs (Thermo Fisher, Waltham, MA, USA. Catalog #18374058). Amplification of the 5′ ends of dsRNAs using RACE used the specific primers GSP1: 5′-ACACCATTGTTAGTACG-3′ and 5′-GAATATACCAGTTGAGG-3′ and GSP2: 5′-GAAGATGATCCACCAACAATAACAGG-3′ and 5′-AGTGGGAAAGGGCAATGTATGG-3′. Amplification of cDNAs from PmV-L-A dsRNAs by reverse transcriptase PCR (RT-PCR) was carried out with the following primer combinations: Reaction 1: 5′-TCCAGTCAATGCTGATAGAGG-3′ and 5′-AGCGGAGCTTCAATACCTGA-3′. Reaction 2: 5′-GCTTATTCAGAGGGGTGGTG-3′ and 5′-AACTGAGCCACCCGAGAATA-3′. Reaction 3: 5′-CGTGGCAGTCAAAAGAAA-3′ and 5′-AACTGAGTCGCACACCCAAT-3′. Reaction 4: 5′-TCGCGATGTTTAGGTGTGAA-3′ and 5′-GTGGTATGCCGACGAATTTT-3′.

### 2.3. Phylogenetic Analysis of PmV-L-A

Gag and Pol protein sequences, from PmV-L-A and its homologs, were aligned using MUSCLE and were inspected manually for accuracy. MEGAX was used to determine each dataset’s appropriate amino acid substitution matrix. MEGAX and PhyML were used to create phylogenetic models of the different viral amino acid sequences [32,33]. For the MEGAX analysis, the maximum likelihood and neighbor-joining methods were used with 1000 bootstrap replicates. PhyML parameters included the program estimating the proportion of invariable sites from the aligned dataset, using the BEST tree topology estimation, a random starting tree, and 1000 bootstrap replicates. Other members of the totivirus subgroups I-B, I-C, and I-D were used as outgroups, compared to the totiviruses of subgroup I-A [34].

### 2.4. Molecular Modeling of the Gag and Pol Proteins of PmV-L-A

AlphaFold2 version 1.2 was used to build 3-D protein structure models with the amino acid sequences derived from the *gag* and *pol* genes [35]. Amino acid sequences were used as an input, with default AlphaFold2 model building parameters (msa_mode: MMseqs2 (uniref + environment); model_type: auto; pair_mode: unpaired + paired; num_recyles: 3). Five Gag models were generated via similar local distance difference test (LDDT) values per residue (Appendix A). AlphaFold2 could only predict one model for the Pol proteins because of their length and low similarity. This resulted in the best-predicted protein model for PmV-L-A Gag, PmV-L-A Pol, ScV-L-A Pol, TdV-LABarr1 Pol, and TAV-1 Pol amino acid sequences. Before energy minimization, the catalytic site Mg^2+^ ion of each RdRP structure was identified. This was then transferred to the predicted model structure via rigid-body backbone alignment to the West Nile virus, RdRP (PDB ID: 2HFZ), using the PyMol visualization software package. Energy minimization was carried out for each model, using the standard protocol described in our previous study [36]. Briefly, each model was placed in a dodecahedron solvent box, and a 10 Å TIP3P water layer was added to solvate the protein model. Na^+^ and Cl^−^ ions, at a concentration of 0.15 M, were then added to the water layers to maintain charge neutrality. The AMBER99SB*-ILDNP force-field parameters were used for the protein and ions. Each system was subjected to energy minimization using the steepest descent algorithm for 10,000 steps with the GROMACS package [37]. To assess the quality of the structural models after energy minimization, stereochemical checks were performed using the SWISS-MODEL structure assessment tool (https://swissmodel.expasy.org/ accessed: 6 November 2021). The final Gag and Pol model structures were analyzed and compared to related viral proteins using the PyMol visualization software package. 

## 3. Results

### 3.1. Purification and Digestion of Double-Stranded RNAs from P. membranifaciens

Our group’s previous identification of *P. membranifaciens* killer yeasts resulted in screening five different strains for the presence of totiviruses and M satellite dsRNAs, using cellulose chromatography [13]. This approach identified a putative dsRNA species within the *P. membranifaciens* strain, NCYC333 (syn. ATCC 36908, CBS 7374), isolated from draught beer in the United Kingdom and deposited in the National Collection of Yeast Cultures in 1953 [12]. The dsRNA showed electrophoretic mobility suggestive of a totivirus (Figure 1, lane 1). The purified nucleic acids were incubated with DNase I or RNase III to confirm that this discrete band was either dsRNA or DNA. The nucleic acid was wholly digested in the presence of RNase III and confirmed that it was a dsRNA molecule (Figure 1, lane 2). 

### 3.2. Determining the Nucleic Acid Sequence of the dsRNA from P. membranifaciens

With the confirmation that *P. membranifaciens* NCYC333 was host to dsRNAs, they were then purified and subjected to poly(A) tailing and cDNA synthesis. The cDNAs derived from the purified dsRNAs were then analyzed by Illumina short-read DNA sequencing [13]. Specifically, the assembly of 42,960 polished Illumina sequence reads revealed five contigs that were more than 1000 bp in length. Blastx analysis identified four contigs related to totiviruses of yeasts (Figure 2B, red dots). Assembly and analysis revealed that the dsRNA species was the genome of a totivirus similar to those previously described in yeasts of the *Saccharomycetaceae* (Figure 2A). Primer pairs were designed to confirm this genome organization by RT-PCR and Sanger sequencing (Figure 2C). The terminal ends of the dsRNA molecule were determined using 5′ RACE and confirmed that the total length of the dsRNA was 4579 bp, which is consistent with a totivirus genome (Genbank accession number OL687555). The sequence analysis of the dsRNA identified an open reading frame of 680 amino acids from nucleotide 28 to 2070 (Figure 2A). The analysis of this amino acid sequence by blastp found relatedness to the Gag protein of totiviruses from yeast species of the *Pichiaceae* and *Saccharomycetaceae* (such as Ambrosiozyma totivirus A (AkV-A; 56% identity), Saccharomyces cerevisiae virus L-A (ScV-L-A; 41% identity), and Torulaspora delbrueckii virus L-A (TdV-L-A; 45% identity)). The Gag protein also contained a conserved histidine at position 154 that is critical for the cap-snatching activity of ScV-L-A [28,38]. Analysis of the 3′ end of the *gag* gene revealed a slippery −1 ribosomal frameshift site sequence (5′-GGGUUU-3′), with a downstream stem-loop (Figure 2D). This frameshift would enable the creation of a Gag-Pol fusion protein of 1522 amino acids that is 49% and 50% identical to ScV-L-A and TdV-L-A, respectively. Other characteristic RNA secondary structures were also evident at the 3′ terminal end of the RNA, including stem-loops characteristic of totivirus replication and packaging signals (Figure 2A and 2D). The latter contained a typical 5′ A-bulge followed by an 11-nucleotide stem-loop observed in other totiviruses and satellite dsRNAs (Figure 2D) [39]. As the first dsRNA virus of the species, we named the new virus Pichia membranifaciens virus L-A (PmV-L-A).

### 3.3. Phylogenetic Analysis of PmV-L-A

To determine the evolutionary relationship of PmV-L-A to other totiviruses, blastp was used to identify homologs of PmV-L-A Gag and Pol proteins. Homologous proteins were mainly from viruses of the genus *Totivirus* in subgroup I-A. Also included were representative viral sequences from the other group-I subgroups (B, C, and D) [34]. Results were filtered to only include those proteins that were > 90% of the length of the PmV-L-A Gag and Pol proteins. Multiple sequence alignments were created with MUSCLE. These alignments were used as inputs for phylogenic analysis, with the neighbor-joining and maximum likelihood methodologies implemented by MEGAX and PhyML [32,33]. In all phylogenic models, PmV-L-A Gag and Pol were placed within the group I-A totiviruses and as outgroups to the *Saccharomycetaceae* virus proteins (Figure 3 and Appendix A). This placement was consistent with *P. membranifaciens* being from the *Pichiaceae* family [41]. PmV-L-A Gag was most closely related to the Gag protein from the incomplete genome of Ambrosiozyma totivirus A (AkV-A) identified within the yeast *Ambrosiozyma kashinagicola,* isolated in Japan (Genbank accession number MK231133.1). Both *P. membranifaciens* and *A. kashinagicola* are classified within the *Pichiaceae* family of yeasts. PmV-L-A proteins appeared to share a common ancestor with another fungal totivirus (tuber aestivum virus 1) and a hypothetical protein in the genome of *Scheffersomyces stipitis* CBS6054 of the CUG yeast clade that includes opportunistic human pathogens, such as *Candida auris* and *Candida albicans* [41]. PmV-L-A proteins and their homologs formed a monophyletic group separate from the subgroup B, C, and D totiviruses associated with fungi and plants (Figure 3).

### 3.4. Structural Modeling of the Gag and Pol Proteins of PmV-L-A

AlphaFold2 was used to predict 3-D structural models to determine whether the Gag and Pol proteins from PmV-L-A were similar in tertiary structure to the models of other viral proteins [35]. AlphaFold2 predicted five 3-D models for the Gag amino acid sequence, while it could only predict a single 3-D model for each Pol amino acid sequence. The LDDT per-residue score selected the best model out of the five predicted Gag models. However, the LDDT score was highly similar for each Gag model, so the first model was selected for further assessment (Appendix A). After generating the models, the structures were subjected to energy minimization using the GROMACS software package. Energy-minimized structural models were then used as inputs to carry out stereochemical checks to assess the quality of the predicted models (Appendix A). Molprobity scores, which combine the clash score, rotamer, and Ramachandran evaluations, were less than 1.4 Å for all computational models, suggesting the good overall quality of the predicted models (Appendix A). 

The final selected model of the PmV-L-A Gag protein was highly similar to the crystal structure model of ScV-L-A Gag (PDB: 1M1C) [42]. Overlaying the two structures revealed a root mean squared deviation (RMSD) of only 1.7 Å, using PDBeFold. In addition, 91% of the secondary structure of the ScV-L-A Gag was identified in the model of PmV-L-A, despite an amino acid identity of only 41% (Appendix A). The modeled Gag also had 95% amino acid residues in the Ramachandran-favored region (Appendix A). In both models, the catalytic histidine 154 was conserved and positioned on the outer surface of Gag, located at the tip of a surface trench formed from four loops (Figure 4A). 

There are no structural models of totivirus polymerases, and they have a less than 20% sequence identity compared to other RdRPs with known structures. Therefore, AlphaFold2 and energy minimization were used to create structural predictions of the PmV-L-A, ScV-L-A, TdV-LABarr1, and TAV-1 Pol proteins. The predicted RdRP structures matched those of other viral RdRP proteins when using PDBeFold. The longest structural match with PmV-L-A Pol was with the VP1 of rotavirus SA11 (PDB 2R7T; group III dsRNA virus, family *Reoviridae* [43]) with 488/866 aligned amino acids, 9.2% identical residues, and an RMSD of 3.96 Å. Similar results were obtained for all modeled totivirus polymerases, which reflects their apparent common ancestry and amino acid conservation (> 49% identity and > 63% similarity) (Appendix A). The modeled Pol proteins also had 90.20–92.89% of amino acid residues in the Ramachandran-favored region (Appendix A). Using the crystal structure model of the rotavirus VP1 as a guide, it was possible to identify three distinct domains in the totivirus polymerases. The core catalytic domain (amino acids 310–697) was sandwiched between an N-terminal domain (amino acids 1–309) and a C-terminal “bracelet” domain (amino acids 698–869) (Figure 4B). The structure of the totivirus C-terminal bracelet domain encircled a pore that, in VP1, is used as an exit for the RNA template during replication. This domain appeared smaller in totiviruses than in VP1 (172 vs. 392 amino acids), with a single predicted antiparallel β sheet and seven α helices (Figure 4B). All the totivirus Pol proteins contained a central cavity, with four entry channels positioned similarly to VP1 (Figure 4C) [44]. Electrostatic maps indicate that these channels are positively charged, with apparent tracts of basic residues that lead to these entry points (Figure 4C). 

The catalytic domain of the modeled RdRPs adopted tertiary structures characteristic of a closed right hand, with palm, fingers, and thumb subdomains (Figure 4D). As with VP1, the palm was comprised of an antiparallel β sheet, supported by three α helices. The thumb consisted of three α helices that interacted with the loops of the fingers, enclosing the central cavity. These subdomains contained seven canonical motifs (from the N-terminus, G, F, A, B, C, D, and E) with conserved residues important for RNA polymerization by RdRP proteins (Appendix A). Our models confirmed that these motifs were positioned within the central cavity of the RdRP. Motif F and G formed part of the fingers subdomain, with motif F interacting with motif E of the palm subdomain to close the catalytic domain. The motifs A, B, C, and D were found in the palm subdomain and were positioned to facilitate catalysis by metal-ion coordination (motifs A and C) and nucleotide triphosphate-binding (motif B). Additional conserved motifs have been previously identified within the RdRPs of totiviruses and have been named motif 1 and motif 2, but their function has not yet been defined [45]. In our models, motif 1 interacted with motif F in the fingers. Motif 2 was within the fingers domain and was exposed to the inner catalytic cavity, where it cradled motif G in a sharp 180-degree turn, aided by a sequence of small amino acids (PGGS) (Appendix A). Overall, these models are consistent with the known structures of Gag and Pol and provide new insights into the structure of the assembled PmV-L-A capsid and the mechanism of totivirus RNA replication.

## 4. Discussion

This article reports the first identified dsRNA virus found in the yeast *P. membranifaciens*. The virus is a new species within the *Totiviridae* family. Although it has diverged from other previously described yeast totiviruses, the sequence and structural conservation exist in viral RNAs and proteins. The relatedness of PmV-L-A to other yeast totiviruses is consistent with the ancestry of the yeast host species. Specifically, PmV-L-A is most closely related to the Ambrosiozyma totivirus A (AkV-A) virus identified within *Ambrosiozyma kashinagicola*, a member of the *Pichiaceae*, like *P. membranifaciens*. PmV-L-A was also an outgroup to the totiviruses found in yeasts of the *Saccharomycetaceae* (i.e., ScV-L-A from *S. cerevisiae* and TdV-LABarr1 from *T. delbrueckii*). The relatedness of these viral species implies the co-evolution of PmV-L-A with *P. membranifaciens.* This is consistent with yeast totiviruses being transmitted via mating and not via an extracellular transmission mode. However, we have shown that not all strains of *P. membranifaciens* host dsRNAs, suggesting that they have lost their totiviruses or that the strain NCYC333 is uniquely infected [13]. Interestingly, *P. membranifaciens* contains the genes that constitute an active RNAi system, which might be expected to prevent viral infection [46,47]. It remains to be determined if the Dicer-like and Argonaute-like proteins of *P. membranifaciens* are functional. Alternatively, PmV-L-A could interfere with the RNAi antiviral system of fungi, as has been shown for other mycoviruses [48,49,50]. *P. membranifaciens* has potentially hosted totiviruses more closely related to viruses found within *Saccharomyces* yeasts, as manifested by genome-integrated sequences [51]. The presence of these genetic fossils would suggest the possibility of horizontal transmission of viral dsRNAs between yeasts, which has also been shown by phylogenetic analysis of mycoviruses and satellite dsRNAs from different yeast species [13,51,52].

The positioning of RNA secondary-structure elements in the genome of PmV-L-A would indicate that they are functionally equivalent to those first described and characterized in ScV-L-A. Although there is only a 54% nucleotide sequence identity between PmV-L-A and ScV-L-A, the similarity of the RNA secondary structures demonstrates conserved strategies for packaging, replication, and frameshifting. For example, the slippery sequence for the −1 frameshift in PmV-L-A is identical to ScV-L-A and other more distantly related totiviruses [34]. Furthermore, the putative packaging signal of PmV-L-A is predicted to form a stem-loop, with the essential A bulge and the G and C nucleotides at the beginning and end of the loop [39]. The first 213 amino acids of the Pol protein are required to encapsidate ScV-L-A RNA transcripts, which corresponds to the N-terminal domain of Pol [31]. Our structural model shows that this N-terminal domain is distinct from the catalytic and bracelet domains and is likely necessary for stabilizing and enclosing the catalytic core of the polymerase.Consistent with its role in packaging, the N-terminal domain has large surface tracts of positive charge that could help to bind the stem-loops of the viral RNAs. 

The computational modeling of the totivirus Pol proteins has shown that the domain organization is consistent with the RdRP subdomains of the palm, fingers, and thumb of other RNA viruses, particularly the group III dsRNAs viruses of the *Reoviridae* (including rotaviruses and reoviruses). The importance of conserved catalytic motifs A–G, common to many RdRPs, has been previously determined for ScV-L-A Pol. We now show that these motifs are positioned similarly in the modeled totivirus RdRPs [45,53]. One interesting exception is that the conserved totivirus-specific motif 1 is placed to block the access of motif G to the central cavity. A recent study suggested that motif G controls RNA translocation, with small amino acids required to form a structural loop [54]. In our models of Pol, motif G is buried by motif 1, which is positioned to face the central cavity. Instead of motif G, motif 1 appears to form a structural loop with small amino acids that could represent an alternative mechanism for regulating RNA translocation by totiviruses.

As with the RdRPs of the *Reoviridae*, the PmV-L-A Pol structural model is permeated by four positively charged pores that would enable strands of RNA to enter and exit the polymerase and the diffusion of small molecules and ions into the catalytic core. The structural similarity to the four-tunnel RdRPs of *Reoviridae* would suggest that totivirus Pol proteins use the same mechanism for the ingress and egress of RNAs (genomic dsRNAs and ssRNAs). This provides a case for the use of yeast totiviruses as a model system to study the mechanism of viral RNA replication. In addition, the discovery of totiviruses in animals [22,23] would further motivate more detailed studies of these viruses, as they could have a significant economic impact on commercial shrimp fisheries [23] and perhaps lead to future disease outbreaks. 

## Figures and Tables

**Figure 1 viruses-14-00594-f001:**
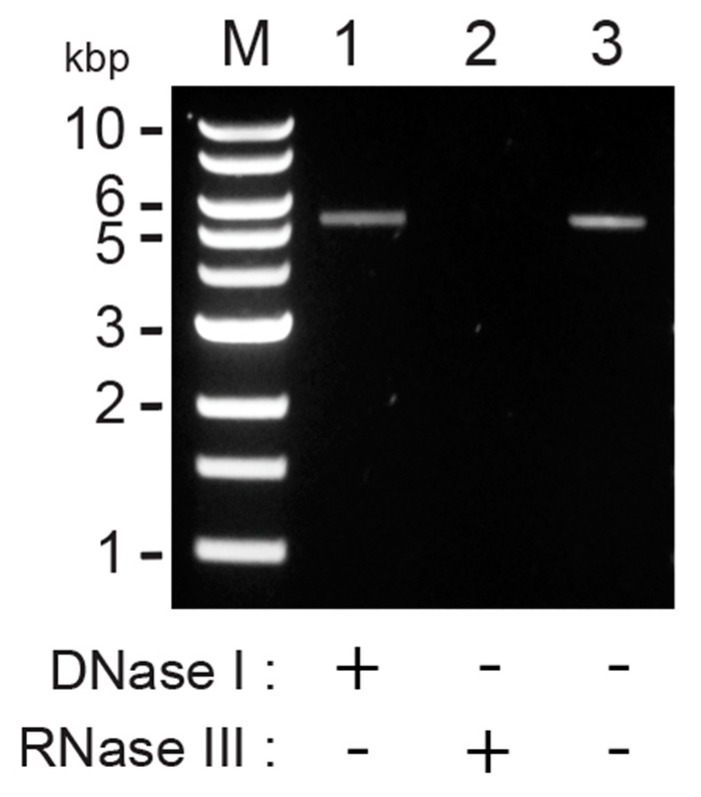
Extrachromosomal nucleic acids within *P. membranifaciens* are dsRNA molecules. Agarose gel electrophoresis of cellulose column-purified nucleic acids from *P. membranifaciens* NCYC333 (lane 3), treated with DNase I (lane 1) or ShortCut^®^ RNase III (lane 2). M indicates a DNA molecular weight marker, with the corresponding sizes labeled.

**Figure 2 viruses-14-00594-f002:**
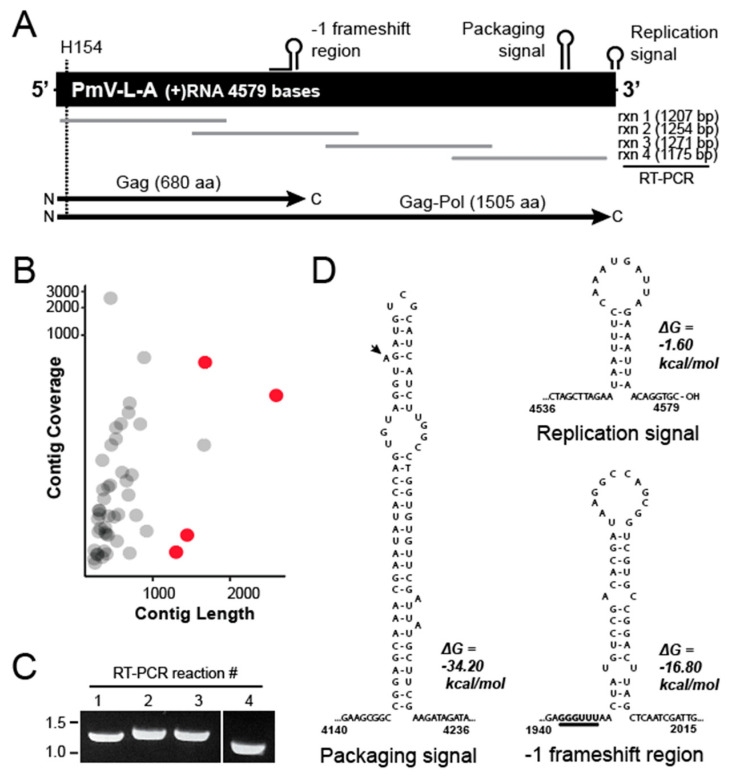
A dsRNA virus within *P. membranifaciens* has the genomic organization of a totivirus. (**A**) Schematic overview of the organization of the PmV-L-A genome, including the position of an RNA secondary structure, ORFs, and a conserved amino acid (H154) required for the cap-snatching activity of Gag. Rxn 1-4 show the position of RT-PCR products used to confirm the genome sequence of PmV-L-A by Sanger sequencing. (**B**) Sequence contigs after de novo assembly are represented by contig coverage and length. Blastx analysis enabled the identification of four contigs related to yeast totiviruses (red). (**C**) Products of RT-PCR, using overlapping primer pairs to confirm the assembled sequence of PmV-L-A. (**D**) RNA secondary structure predictions were performed using the mFold server [40].

**Figure 3 viruses-14-00594-f003:**
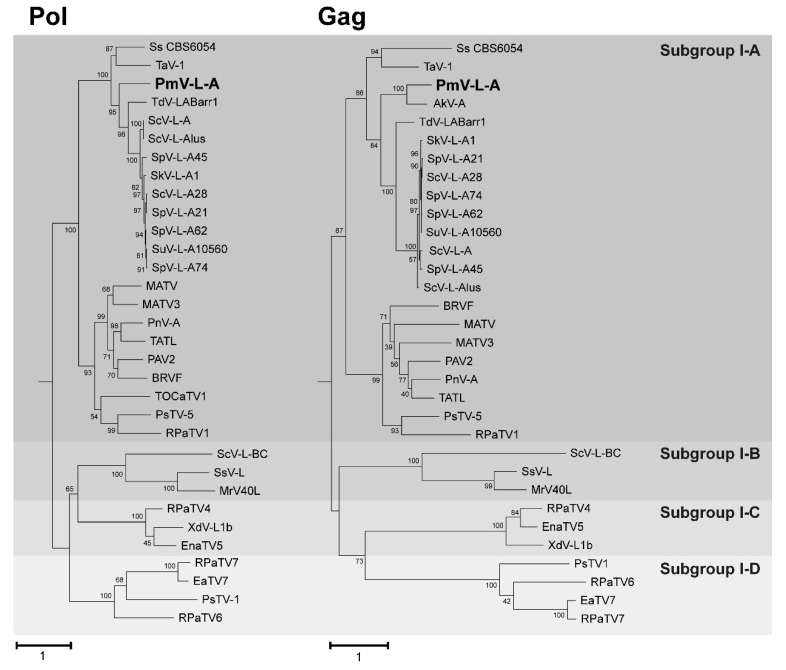
Phylogenetic analysis of the Gag and Pol proteins encoded by PmV-L-A. Rooted maximum likelihood phylogenic models were constructed using PhyML with Pol (**left**) and Gag (**right**) proteins from PmV-L-A (bold) and related totiviruses. The LG model with a gamma distribution (G) and invariable sites (I) was the substitution matrix that best fits these datasets. The numbers at each node are the bootstrap values from 1000 iterations. The scale bar represents the distance of one amino acid substitution per site. The amino acid sequences in the phylogeny are from the viruses listed in Appendix A.

**Figure 4 viruses-14-00594-f004:**
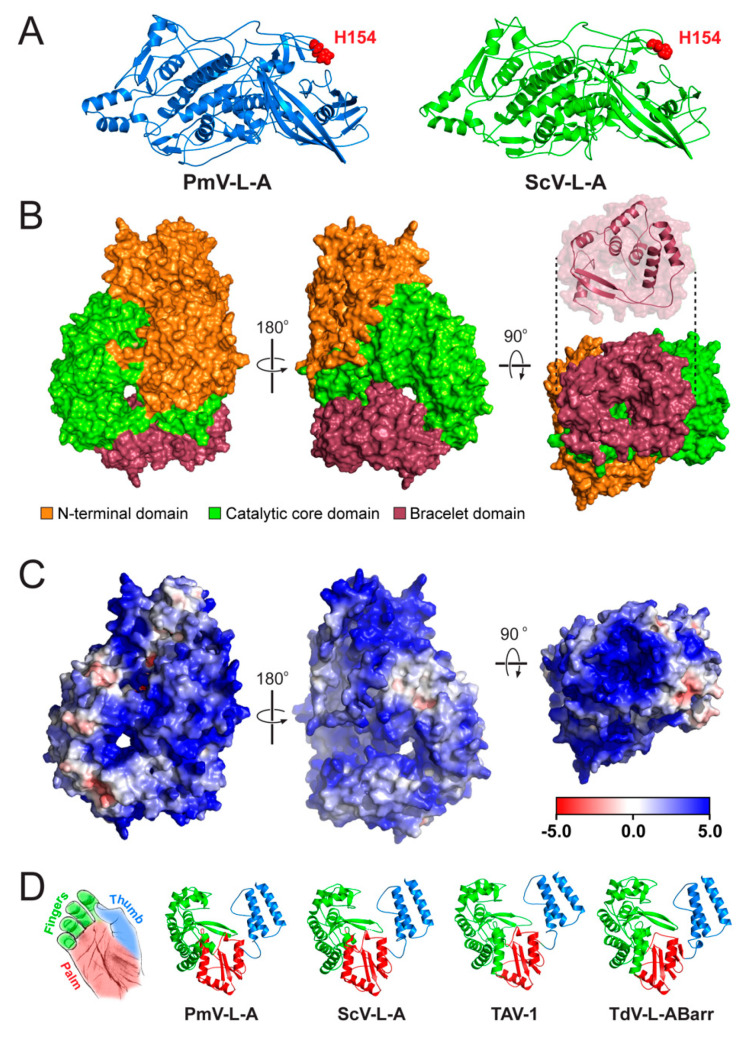
Structural models of the PmV-L-A Gag and Pol proteins and comparison to other totivirus polymerases. The AlphaFold2 server and energy minimization were used to generate molecular models of proteins encoded by PmV-L-A and related totiviruses. (**A**) Cartoon models of the Gag protein of PmV-L-A, compared to the crystal structure model of the Gag protein of ScV-L-A. The catalytic histidine (H154) is depicted in red in each model. (**B**) Surface model of the PmV-L-A Pol protein, representing the N-terminal domain (orange), catalytic domain (green), and bracelet domain (burgundy). The bracelet domain is also depicted as a cartoon to illustrate the channel structure more clearly. (**C**) Surface electrostatic model of the PmV-L-A Pol, using an adaptive Poisson Boltzmann solver (APBS). (**D**) Cartoon representation of the catalytic domain of four modeled totivirus polymerases, showing the finger, thumb, and palm subdomains.

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
