# Peer review of "The Characterization of a Novel Virus Discovered in the Yeast Pichia membranifaciens"

_viruses, 2022, doi:10.3390/v14030594_

Round 1

Reviewer 1 Report

The paper from Mark D. Lee and co-authors presents the identification of a novel member of the Totiviridae isolated from P. membranifaciens. In my opinion the information contained in the paper are of interest, but the authors need to supply substantial adjustment prior to be published.

Comment 1: Did the author detect PmV-L-A in other P. membranifaciens strains? What is the phenotype of P. membranifaciens strain contained PmV-L-A? The author should compare the strain NCYC777 with virus-free stains.

Comment 2: The author needs to purify the virions and show the electron micrograph of PmV-L-A particles.

Comment3: Totivirus is phylogenetically separated into two groups, I and II. And the members of group I are further divided into four subgroups, A to D (Kondo et al., 2016). The reviewer suggests that the author should rebuild the phylogenetic trees showing group/subgroups. (Kondo H , Hisano S , Chiba S , et al. Sequence and phylogenetic analyses of novel totivirus-like double-stranded RNAs from field-collected powdery mildew fungi[J]. Virus Research, 2016.)

Comment 4: A DNA marker should be included in Figure 1.

Author Response

REVIEWER 1:

Comment 1: Did the author detect PmV-L-A in other P. membranifaciens strains? What is the phenotype of P. membranifaciens strain contained PmV-L-A? The author should compare the strain NCYC777 with virus-free stains.

            RESPONSE: We previously published an analysis of the dsRNAs within five strains of P. membranifaciens using cellulase chromatography (see Fredericks et al. PLoS Genetics 2021). We have now highlighted this manuscript to draw attention to this fact: Line 146 “The previous identification of P. membranifaciens killer yeasts by our group had resulted in the screening of five different strains for the presence of totiviruses and M satellite dsRNAs using cellulose chromatography [13].”

Comment 2: The author needs to purify the virions and show the electron micrograph of PmV-L-A particles.

            RESPONSE: We have provided extensive data that detail the genomic organization of the newly discovered virus, including its genome sequence, the direct visualization of its genome, molecular modeling of its proteins, and its close similarity to other yeast totiviruses. As this manuscript is a short communication, and we do not have the facilities or the current funds for TEM analysis, we believe that we have provided sufficient evidence of the new totivirus in line with other similar publications in mdpi Viruses.

Comment3: Totivirus is phylogenetically separated into two groups, I and II. And the members of group I are further divided into four subgroups, A to D (Kondo et al., 2016). The reviewer suggests that the author should rebuild the phylogenetic trees showing group/subgroups. (Kondo H , Hisano S , Chiba S , et al. Sequence and phylogenetic analyses of novel totivirus-like double-stranded RNAs from field-collected powdery mildew fungi[J]. Virus Research, 2016.)

            RESPONSE: We have included representative totiviruses from all four subgroups of group I totiviruses. The phylogenetic model is annotated to clearly show these subgroups. We have also referenced Kondo et al. to provide some context for this organization.

Comment 4: A DNA marker should be included in Figure 1.

            RESPONSE: Done

Reviewer 2 Report

The manuscript presents the discovery and complete molecular characterization of a totivirus infecting the yeast Pichia membranifaviens and the phylogenetic relationships with other viruses. The authors also showed the structural models of the Gag and Pol viral proteins.

Comments:

Pag1 line 36. The authors mention that the majority of mycoviruses have linear dsRNA genomes. This information is outdated, nowadays, multiple works indicate that the majority of mycoviruses have genomes of single-stranded positive sense RNA viruses.

Pag3 line 100. For the phylogenetic analysis the bootstrap replicates were 100 or 1000? The normal is to use 1000, maybe is a mistake in the text.

Pag3 line 147. Add the next sentence: “confirming that was a dsRNA element” after “…in the presence of RNase III (Figure 1, lane 2)”

Pag4 line 156. The information of Illumina short-read DNA sequencing should be included and described in Materials and Methods, in 2.2. And explain if this method was used for viral detection before the complete sequencing of the viral genome by Sanger sequencing.

Pag5. Figure 2C, include marker in the gel. Total length of the dsRNA was 4578 bp (line 164) or 4579 (Figure 2A).

Pag6. Add a symbol of plant, fungi or yeast in the phylogenetic tree of Figure 3 to see in one single look the organization. Maybe the authors should repeat the phylogeny including more sequences from other totiviruses infecting other fungi or oomycete to determine if the organization of the groups is similar.  Include it in Figure S1, and change phylogenetic trees of Gag to the right and trees of the Pol to the left, to be in the same order as Figure 3.

Pag7. Indicate the amino acid sequence identity of the proteins of the different mycoviruses used in the structural modeling, both, Gag and Pol.  Include a figure with the alignment of these proteins indicating the typical motifs. Indicate in Figure 4 with arrows, for example, what is described in the text.

Pag9. Line 302. Different strains may host different viruses or no viruses. How many strains of the yeast have been analyzed to find viruses? Maybe this paragraph is too speculative.

Author Response

REVIEWER 2:

The manuscript presents the discovery and complete molecular characterization of a totivirus infecting the yeast Pichia membranifaciens and the phylogenetic relationships with other viruses. The authors also showed the structural models of the Gag and Pol viral proteins.

Comments:

Pag1 line 36. The authors mention that the majority of mycoviruses have linear dsRNA genomes. This information is outdated, nowadays, multiple works indicate that the majority of mycoviruses have genomes of single-stranded positive sense RNA viruses.

            RESPONSE: On line 40 we have altered this sentence to avoid the misrepresentation of the prevalence of dsRNA viruses in fungi.

Pag3 line 100. For the phylogenetic analysis the bootstrap replicates were 100 or 1000? The normal is to use 1000, maybe is a mistake in the text.

            RESPONSE: We have now increased the number of bootstrap replicates to 1000 and updated the phylogenetic trees accordingly.

Pag3 line 147. Add the next sentence: “confirming that was a dsRNA element” after “…in the presence of RNase III (Figure 1, lane 2)”

            RESPONSE: Done

Pag4 line 156. The information of Illumina short-read DNA sequencing should be included and described in Materials and Methods, in 2.2. And explain if this method was used for viral detection before the complete sequencing of the viral genome by Sanger sequencing.

            RESPONSE: The virus was first discovered because of screening five strains of P. membranifaciens as we have now clarified on Line 146. We have added a clearer reference to our previously published work and methodologies in the manuscript Fredericks et al. PLoS Genetics 2021 that clearly detail the exact protocol used for dsRNA sequencing. Line 164 “With the confirmation that P. membranifaciens NCYC333 was host to dsRNAs, they were then purified and subjected to poly(A) tailing and cDNA synthesis. The cDNAs derived from the purified dsRNAs were then analyzed by Illumina short-read DNA sequencing [13].”

Pag5. Figure 2C, include marker in the gel. Total length of the dsRNA was 4578 bp (line 164) or 4579 (Figure 2A).

RESPONSE: Done

Pag6. Add a symbol of plant, fungi or yeast in the phylogenetic tree of Figure 3 to see in one single look the organization. Maybe the authors should repeat the phylogeny including more sequences from other totiviruses infecting other fungi or oomycete to determine if the organization of the groups is similar.  Include it in Figure S1, and change phylogenetic trees of Gag to the right and trees of the Pol to the left, to be in the same order as Figure 3.

            RESPONSE: To the phylogenetic analysis we have now added more Gag and Pol sequences from other more distantly related totiviruses to better show the relatedness of type I totiviruses. We have also added annotations to show that members of group I Totivirus are phylogenetically separated into subgroups referencing Kondo H , Hisano S , Chiba S , et al. Sequence and phylogenetic analyses of novel totivirus-like double-stranded RNAs from field-collected powdery mildew fungi[J]. Virus Research, 2016.).

Pag7. Indicate the amino acid sequence identity of the proteins of the different mycoviruses used in the structural modeling, both, Gag and Pol.  Include a figure with the alignment of these proteins indicating the typical motifs. Indicate in Figure 4 with arrows, for example, what is described in the text.

RESPONSE: As suggested by the reviewer, we now include amino acid similarity and identity scores for both Gag and Pol for the four totiviruses that we have modeled in our study. We also now include an alignment of the RdRp sequences to enable us to highlight the similarities in the major catalytic residues of these proteins that are discussed in the body of the text. Both of these additions constitute a new supplemental figure S3.

Pag9. Line 302. Different strains may host different viruses or no viruses. How many strains of the yeast have been analyzed to find viruses? Maybe this paragraph is too speculative.

RESPONSE: From our previous work we have analyzed five different strains of P. membranifaciens (we have included more details on this work and a reference (Line 146). In the discussion section we discuss the two potential origins of PmV-L-A (horizontal vs vertical transmission) (Lines 314-316).

Round 2

Reviewer 1 Report

I am satisfied with the paper in its current form.

Reviewer 2 Report

The authors answered my questions and accepted my suggestions. I don't have further comments.